# Comprehensive mutational scanning of EGFR reveals TKI sensitivities of extracellular domain mutants

Tikvah K. Hayes[1,2,10], Elisa Aquilanti[1,2], Nicole S. Persky[2,3,11], Xiaoping Yang [3], Erica E. Kim[1], Lisa Brenan[2], Amy B. Goodale[3], Douglas Alan[3], Ted Sharpe[4], Robert E. Shue[1,2], Lindsay Westlake[2], Lior Golomb[1,2], Brianna R. Silverman[1], Myshal D. Morris[5], Ty Running Fisher[5], Eden Beyene[5], Yvonne Y. Li [1,2], Andrew D. Cherniack [1,2], Federica Piccioni [3,12], J. Kevin Hicks[6], Andrew S. Chi[7], Daniel P. Cahill[7], Jorg Dietrich[8], Tracy T. Batchelor[9], David E. Root [3], Cory M. Johannessen [2,13] & Matthew Meyerson [1,2] ✉

The epidermal growth factor receptor, EGFR, is frequently activated in lung cancer and glioblastoma by genomic alterations including missense mutations. The different mutation spectra in these diseases are reflected in divergent responses to EGFR inhibition: significant patient benefit in lung cancer, but limited in glioblastoma. Here, we report a comprehensive mutational analysis of EGFR function. We perform saturation mutagenesis of EGFR and assess function of ~22,500 variants in a human EGFR-dependent lung cancer cell line. This approach reveals enrichment of erlotinib-insensitive variants of known and unknown significance in the dimerization, transmembrane, and kinase domains. Multiple EGFR extracellular domain variants, not associated with approved targeted therapies, are sensitive to afatinib and dacomitinib in vitro. Two glioblastoma patients with somatic EGFR G598V dimerization domain mutations show responses to dacomitinib treatment followed by within-pathway resistance mutation in one case. In summary, this comprehensive screen expands the landscape of functional EGFR variants and suggests broader clinical investigation of EGFR inhibition for cancers harboring extracellular domain mutations.

The *epidermal growth factor receptor*, or EGFR, is a transmembrane receptor tyrosine kinase (RTK), that like many other RTKs, is commonly mutated in cancer. EGFR alterations are observed in ~5% of all cancer patients, 14% of patients with non-small cell lung cancer, and 26% of patients with glioblastoma[1–5]. The use of targeted panel sequencing has helped identify multiple hotspot activating alterations in *EGFR*-mutant cancers, most notably missense mutations in EGFR at amino acids A289, G719, and L858, although activating alterations in EGFR can also include in-frame insertions and deletions, truncations, fusions, and amplifications[6–8]. When assessing

EGFR missense mutations, glioblastomas are generally characterized by extracellular domain mutations, while mutations in *EGFR*-mutant lung cancers generally arise in the kinase domain[4,9,10]. However, there are reciprocal examples where we find EGFR extracellular domain mutations in lung cancer and kinase domain mutations in glioblastoma[1,11,12]. In addition to mutations that have been definitively characterized as oncogenic drivers, other subsets of EGFR missense mutations with unknown functional relevance have been, and continue to be, identified in patients. Variants with no reported function are classified as variants of unknown significance (VUS),

while oncogenic drivers are classified as variants of known significance (VKS).

Given the prevalence of patient-observed EGFR variants, several generations of EGFR tyrosine kinase inhibitors (TKIs) have been, and continue to be, developed[13,14]. Clinical response to EGFR TKIs has varied, where subsets of EGFR variants, like L858, are sensitive to first and third generation TKIs[15–17], while uncommon mutations (G719, S768, and L861) have responded better to second generation TKIs[18]. Furthermore, there are no currently approved therapeutic options for cancers with EGFR variants located outside of the kinase domain. Clinical trials have investigated current EGFR therapeutics for treatment of cancers bearing these non-kinase domain variants[19] and this remains an active area of investigation. Thus, establishing functionality and inhibitor sensitivities for all EGFR missense variants, both for those already observed in patients as well as mutants that have not yet been reported but might have functional impact, could potentially allow patients with non-canonical *EGFR*-mutant cancers to benefit from existing approved EGFR-targeted drugs. This could also stimulate the development of novel EGFR-directed therapies.

Given the importance of the comprehensive assessment of EGFR mutant function, several recent studies have generated analyses of large collections of such mutants[20–22]. One study applied the "mixed-all-nominated-mutants-in-one" approach performing pooled screens of 101 patient-observed EGFR mutations, assayed by cellular transformation and inhibitor sensitivity in murine cell lines[21]. A second study performed an "in vitro screen for activating mutations" to assess the function of ~7,200 EGFR variants, including patient-observed and randomly generated mutants, assessed by growth-factor independence and inhibitor studies in the murine hematopoietic Ba/F3 cell line[22]. Finally, a comprehensive study of EGFR kinase domain variants, that sub-grouped variants based on kinase domain locations, also used the Ba/F3 model to assess inhibitor sensitivity in correlation with structural analysis[20].

These previous studies were relatively comprehensive, but the available technologies did not yet allow the study of all possible EGFR mutations. The development of new methods, such as the MITE (*M*utagenesis by *I*ntegrated *TilE*s) technology[23], has enabled the comprehensive evaluation of protein function, revealing important insights into structure/function relationships and sensitivity to targeted therapies[24–26]. The further development of synthetic site saturation mutagenesis methods based on high-throughput oligonucleotide synthesis now permits complete structure function analysis of all possible substitution mutations, even in large genes[27]. Here, we applied experimental and computational methods for library design and deconvolution that permit the screening of very large variant pools covering 1000s of nucleotides of an open reading frame[28]. These recent technological advancements have positioned the field to comprehensively evaluate and prospectively map protein variant functionality in disease.

In this study, we perform a pooled EGFR variant genetic screen to identify erlotinib-responsive and -nonresponsive variants in human models of lung cancer. To the best of our knowledge, this study represents the largest variant pool and *EGFR* is the largest gene to be screened to date with whole-gene single-pool saturation mutagenesis. Our analyses yield an enrichment of functional EGFR variants in the dimerization, transmembrane and kinase domains, thus expanding the landscape of functional EGFR variants. We then test a subset of patient-observed and poorly characterized variants, finding many of them to be active in the presence of both erlotinib and osimertinib. In this subset, EGFR TKI insensitivity is accompanied by increases in phospho-ERK levels leading to pathway reactivation downstream of the inhibitor block. Several EGFR extracellular domain variants are dacomitinib-sensitive in vitro and we observed clinical response to dacomitinib in patients whose cancers harbor the G598V mutation. Our study provides a systematic and comprehensive description of EGFR variant functionality in models of lung cancer.

## Results

### Characterization of PC9 lung cancer cells as a model for EGFR variant functionality

Previous studies of EGFR mutational function have relied on ectopic expression in murine cell lines of non-lung origin such as Ba/F3, a hematopoietic cell line where EGFR mutants drive interleukin-3 independence, and NIH-3T3, where EGFR mutants lead to transformation as assayed by focus formation and/or growth in soft agar[20–22,29,30]. For this study, we wanted to evaluate and expand EGFR variant function under a more physiological context and therefore developed a lung cancer model system for this purpose. To develop a model to study a saturated library of mutated EGFR proteins in a physiologically context, we validated the use of PC9 cells, a non-small cell lung cancer (NSCLC) cell line, which contains an endogenous EGFR exon 19 deletion. This well-characterized cell line is known to be EGFR-dependent, is sensitive to all generations of EGFR TKIs, and is readily susceptibly to large-scale screening studies[31,32]. The PC9 model has advantages and disadvantages relative to some widely used murine models such as Ba/F3 and NIH-3T3. One advantage is using a more physiological context of a naturally EGFR-dependent lung cancer cell line while a disadvantage is that our ectopic expression system leads to co-expression of two forms of EGFR−the endogenous exon 19 deletion mutant and the exogenously introduced missense mutant form. Another difference, and possible advantage, is that PC9 cells contain the naturally occurring range of somatic passenger mutations of a lung adenocarcinoma.

To validate the PC9 cell line as a model for the assessment of EGFR variant functionality, we first analyzed a set of variants of known significance (VKS) with varying sensitivities to erlotinib, spanning the EGFR protein (Fig. 1a), in functional assays including cell viability and colony formation, as well as biochemical downstream signaling assays. In these assays, mock infection and LacZ expression serve as erlotinib-sensitive controls, while expression of EGFR T790M, with an otherwise WT kinase domain, is an erlotinib-insensitive control. EGFR WT can drive limited growth in the presence of erlotinib in PC9 cells and serves as a baseline for comparison of the drug sensitivity and functional impact of the derived variants. This is consistent with the observation that overexpression of EGFR WT modulates TKI-sensitivity in previous studies[33,34]. We used these controls as standards to characterize the ability of the VKS panel to grow in the presence of erlotinib.

Next, we generated PC9 cell lines with mock infection or expressing LacZ, EGFR WT, EGFR T790M, EGFR extracellular or intracellular domain variants (Fig. 1a and Supp. Fig. 1a). We treated these cell lines with increasing concentrations of erlotinib for 144 h and measured cell viability. We found the controls behaved as expected, where EGFR WT demonstrated reduced erlotinib-sensitivity compared to either mock transduced or LacZ, while EGFR T790M was erlotinib-insensitive[31]. The panel of EGFR extracellular domain variants also displayed reduced sensitivity to erlotinib as shown by a shift in the dose-response curves. As expected, a subset of kinase domain variants (G719A, G719C, G719D, G719S, S768I, and L858R) were inhibited by erlotinib, while others (E709A, L747P, L747S, V769L, G779F, and L861K) were less functionally responsive to erlotinib in this assay (Fig. 1b and Supp. Fig. 1a). Next, a colony formation assay was performed to understand the effects of 200 nM erlotinib treatment after 10 d. The colony formation phenotypes were consistent with the cell viability assay with regard to erlotinib effects (Fig. 1c and Supp. Fig. 1b).

Next, we assessed the consequences of ectopic expression of EGFR VKS, with or without erlotinib inhibition, on downstream signaling in PC9 cells. Overexpression of EGFR VKS was confirmed by total EGFR immunoblotting and also resulted in increased levels of phosphorylated EGFR, but not phospho- or total ERK, or total cyclin B1

(Fig. 1d-g and Supp. Fig. 1c). We also observed increased levels of phosphorylated STAT1 and STAT3 in cell lines expressing extracellular domain VKS (R108K, M277E, A289V, and G598V) (Supp. Fig. 1d). A subset of intracellular domain VKS (E709A and L747P) also displayed

increased levels of STAT1 and STAT3 (Supp. Fig. 1e). It is unclear why baseline expression of EGFR (WT, T790M, R108K, M277E, A289V, E709A, or L747P) in the PC9 cell line led to increased levels of phospho-STAT1 as increased levels of phospho-STAT1 did not appear to

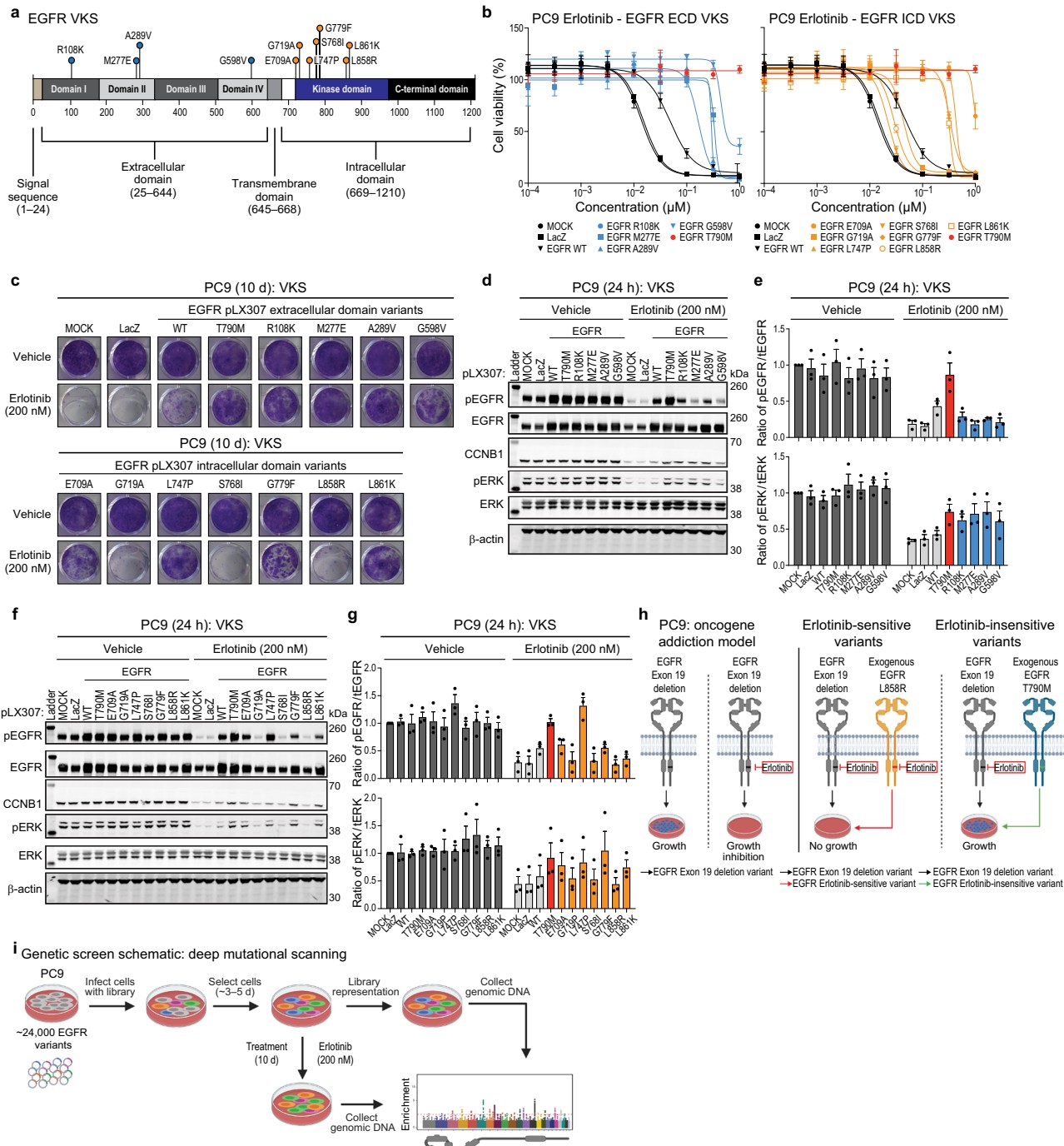

**Fig. 1 | Validation of VKS in PC9 oncogene addiction model and screening schematic. a** EGFR domain schematic showing tested variants of known significance (VKS). **b** PC9 cell lines expressing either LacZ, EGFR WT, EGFR T790M, EGFR extracellular domain variants (R108K, M277E, A289V, or G598V), or EGFR intracellular domain variants (E709A, G719A, L747P, S768I, G779F, L858R or L861K), after 144 h treatment with increasing doses of erlotinib (normalized to vehicle control). A representative experiment is shown, remaining biological replicates are located in Source Data (N = 3). Data are presented as mean values ± SD. **c** Colony formation assay in PC9 cell lines expressing either LacZ, EGFR WT, EGFR T790M, EGFR extracellular domain variants (R108K, M277E, A289V, or G598V), or EGFR intracellular domain variants (E709A, G719A, L747P, S768I, G779F, L858R or L861K)

after a 10 d of treatment with either vehicle (DMSO) or 200 nM erlotinib. A representative experiment is shown (N = 3). **d–g** Representative immunoblots displaying the effect of 200 nM erlotinib after 24 h treatment on PC9 cell lines expressing either LacZ, EGFR WT, EGFR T790M, EGFR extracellular domain variants (R108K, M277E, A289V, or G598V) (**d, e**), or EGFR intracellular domain variants (E709A, G719A, L747P, S768I, G779F, L858R or L861K) (**f, g**) on the levels of both phosphorylated EGFR and ERK and total EGFR and ERK. β-actin immunoblotting was used to determine equivalent loading. Data are presented as mean values ± SEM of biological replicates (N = 3) (**e, g**). **h** PC9 oncogene addiction model. **i** Screening schematic and timeline.

correlate with variant activity in the presence of erlotinib. After 24 h of erlotinib treatment, the positive control (EGFR T790M), EGFR extracellular VKS, and a subset of intracellular VKS (E709A, L747P, L747S, V769L, G779F, and L861K) displayed higher levels of phospho-ERK, but not phospho-AKT, -STAT1, or -STAT3 compared to either mock transduced, LacZ, or EGFR WT (Fig. 1d–g and Supp. Fig. 1d, e).

Increased phospho-ERK levels appeared to be associated with growth in the presence of erlotinib in both the cell viability and colony formation assays. Given this observation, we treated the PC9 VKS expressing cell lines with 20 nM trametinib, alone and in combination with erlotinib (200 nM), to determine whether the panel was dependent on MAPK for growth in the presence of erlotinib. We observed no change in colony formation after 10 d for cell lines treated with trametinib alone, though we observed reduced colony formation, to differing degrees, for all variants including EGFR T790M in cells lines treated with the combination (Supp. Fig. 1f). These data suggest at least a partial MAPK dependency for PC9 erlotinib-insensitive EGFR variants. Combination treatment also resulted in reduced phospho-ERK levels, consistent with a reduction in colony formation (Supp. Fig. 1g, h). These data support previous findings, suggesting EGFR variants are dependent on MAPK signaling for growth[35,36]. Together, these data suggest that the assessment of EGFR variant function in PC9 cells is reliable for analysis of cell growth and survival and for downstream signaling.

### Design and implementation of a screen for erlotinib-insensitive EGFR variants in the PC9 oncogene addiction model

Next, we optimized screening conditions for selection of erlotinib-insensitive EGFR variants in the PC9 cell line. We employed several controls to define the appropriate erlotinib dose and time point. We generated PC9 cell lines expressing WT and mutant EGFR (T790M, L858R, or T790M/L858R) and assessed them by measurement of cumulative population doublings for 15 d. We assessed the effects of erlotinib at 30 nM, 50 nM, 100 nM, and 200 nM, and also included DMSO as a vehicle control. Based on cumulative population doublings, we concluded that 200 nM erlotinib treatment for 10 d would provide a reasonable distinction between PC9 cells expressing EGFR variants with differing responses to erlotinib (Supp. Fig. 2a-e). This leads to the overall scheme for the EGFR variant function assessment in this study (Fig. 1h) (Created with BioRender.com). PC9 cells are dependent on EGFR for growth and survival, and at baseline, are sensitive to erlotinib (Fig. 1h, left panel). When transduced with growth-supporting and erlotinib-sensitive EGFR variants, the PC9 cells do not grow in the presence of erlotinib (Fig. 1h, middle panel); this is also true for non-functional mutants. However, when transduced with growth-supporting and erlotinib-insensitive EGFR variants, the PC9 cells continue to grow in the presence of erlotinib (Fig. 1h, right panel).

To assess the functionality of a broad range of single amino acid substitutions within EGFR, we designed and synthesized a deep mutational scanning library containing approximately 24,000 EGFR variants, comprising almost every substitution of all 20 possible amino acid variants including stop codons for each of the 1210 positions in full-length EGFR, and cloned this variant library into a lentiviral expression vector (Fig. 1i and *Methods*) (Created with BioRender.com). Using a pooled format, we overexpressed the EGFR variant library in the PC9 cell line and selected for cells that incorporated a single variant. Then we split the infected cells into two populations, using one portion to assess initial library representation and the other portion for screening in the presence of erlotinib. We treated the screening plates with 200 nM of erlotinib for 10 d and then isolated genomic DNA and performed next generation sequencing (NGS) to determine variant enrichment (Fig. 1i). NGS analysis after screening revealed a variant library coverage rate of 93% (Supp. Fig. 2f), giving a total estimate of ~22,500 covered variants (Supp. Data 1). As expected, our positive control, EGFR T790M, was amongst the most enriched variants, while our negative control, EGFR L858R, was not enriched as assessed by z-score (Fig. 2a).

### Results of deep mutational scanning of full-length EGFR

We assessed the abundance of each variant in the pooled PC9 cell population following erlotinib treatment. Variants were quantified via enrichment or depletion by calculating the z-score of the $\log_2$(fold change) (LFC) with erlotinib treatment, relative to pretreatment reference abundance (Fig. 2a). While there was inevitable variability in the assay given the depth of coverage attainable, 97.5% of variants were associated with z-scores with an absolute value of <2 and 99% <3. When we assessed the enrichment or depletion of specific variants, very few were deeply depleted, suggesting the rarity of dominant negative mutants at the relative expression levels tested. The most well-characterized EGFR dominant negative is the CD533 truncation, which is missing the last 533 amino acids[37]. Given this, we would expect the corresponding variant to be depleted in our screen. Interestingly, we did not observe depletion for variant R677* (not represented in the input variant library after transduction), though R675* (z-score < −3), was the most depleted truncation (Fig. 2a,b). However, a number of variants were specifically enriched after erlotinib selection, including 559 variants with enrichment z scores >2, among which there were 216 variants with z scores > 3, and 88 variants with z scores > 4 (Supp. Data 2). Of the tested variants, >95% with a z-score >2 have not been previously reported according to our analysis of published comprehensive functional studies[20–22,38] (Supp. Data 2 and Supp. Table 1). We observed an enrichment of variants across the EGFR coding sequence, with the strongest enrichment of variants located in the domains responsible for dimerization, the transmembrane domain, and the kinase domain (Fig. 2a). Looking at specific amino acid substitutions, we observed an enrichment of variants where the normal coding residue was replaced by cysteine in the dimerization domains, mostly notably in domains II and IV (Fig. 2b). These cysteine substitutions could create more disulfide-bonded molecules in regions, like domain II and IV, which are already rich in cysteines[39]. There were several EGFR positions (A289, C311, C333, C582, C595, C620, C624, C628, E709, and L718), where regardless of amino acid substitution, we observed enrichment as measured by the average z-score (Fig. 2c). Interestingly, these residues all correspond to known sites of mutation or to cysteines in the original sequence. Furthermore, we observed domain-specific enrichment where domain II (189-333), domain IV (504-644), the transmembrane domain (645-668), and the C-terminal domain (979-1210) all had average z-scores > 0 (Fig. 2c).

Next, we compared the screening analysis to patient-observed somatic EGFR variants found in GENIE, focusing first on glioblastoma (Fig. 2d) and lung cancers (Fig. 2e) as those cancers commonly harbor EGFR mutations. We observed some overlap with known oncogenic variants (A289, G598, E709, L747, and L861), while identifying and expanding potential functionality for previously uncharacterized variants. As expected, there were several variants, known to be oncogenic (L858), and erlotinib-sensitive, that were not selected for in our PC9 screen (Fig. 2f). Together, our screening results revealed both expected VKS, and unexpected VUS.

### Validation of erlotinib-insensitive EGFR variants of unknown significance (VUS) observed in human cancers

As shown above, the saturation mutagenesis screen identified a large collection of EGFR substitution variants which confer altered erlotinib sensitivity in PC9 cells. For validation, we chose to focus on a subset of erlotinib-insensitive variants that have been observed in sequencing data from human lung cancers and glioblastomas. We selected variants with enrichment z scores > 1.5 that were observed at least 2 times in GENIE, COSMIC, or TCGA mutational data[3,40,41]. Based on this selection, we tested a subset of EGFR extracellular (R222C, S229C, A237Y, T302H, C311R, S447F, C595G, and P644S) and intracellular (T725M, V769M,

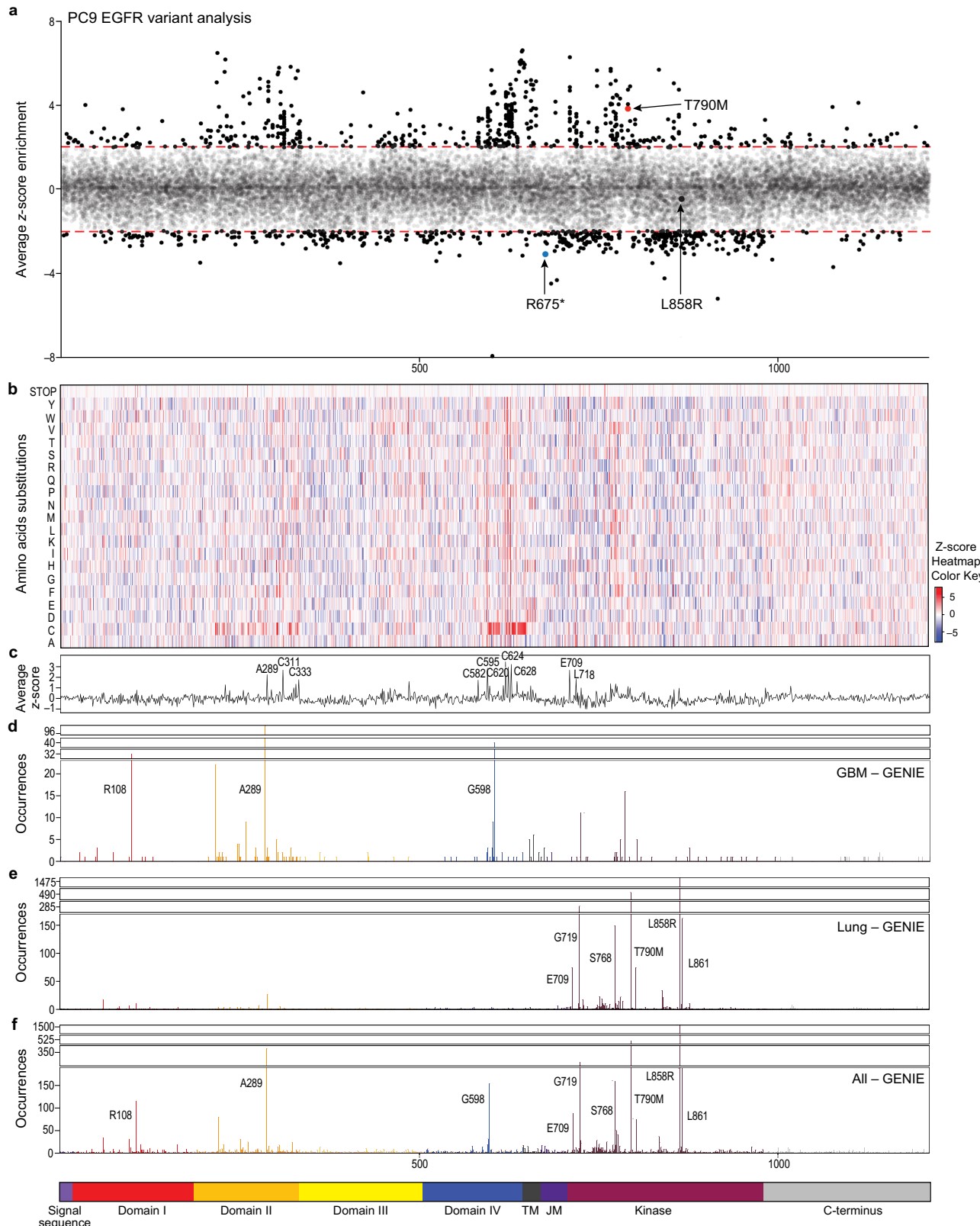

**Fig. 2 | Systematic identification of erlotinib-insensitive EGFR variants using deep mutational scanning. a–f** PC9 cell line stably expressing an EGFR missense variant library was treated for 10 d with 200 nM erlotinib. Alignment of EGFR variant library analysis by z-score enrichment of position (**a**) or amino acid substitution (**b**), average z-score by position (**c**), and EGFR patient observed mutations in glioblastoma (**d**), lung (**e**), and all cancers (**f**) (GENIE). For the EGFR schematic below, TM (transmembrane) and JM (juxtamembrane).

H773R, and V774M) domain variants (Fig. 3a). Of the 12 variants selected for further analysis, 8 (A237Y, T302H, C311R, S447F, C595G, P644S, T725M, and H773R) have not undergone functional analysis beyond clinical identification and in silico assessment based on our current understanding of the literature. Given this, we expressed this panel of EGFR VUS in three EGFR mutant NSCLC cell lines, PC9, HCC4006, and HCC827, and assessed cell viability after 144 h by treating cells with increasing concentrations of erlotinib. Cells expressing the tested EGFR extracellular domain (ECD) variants were less sensitive to erlotinib as seen by a shift in the dose response curve compared to mock transduced, LacZ, and WT expressing cells, though none of the mutants in the panel was as erlotinib-insensitive as the positive control T790M (Fig. 3b). The EGFR P644S variant displayed a phenotype more closely associated with EGFR WT in each of the tested lung cancer models. Cells expressing the EGFR intracellular domain variants were also less sensitive to erlotinib in a cell viability assay compared to cells expressing EGFR WT or the negative controls (Fig. 3c). Next, we performed a colony formation assay to better understand the longer-term effects of expressing the tested EGFR variants. Similar to the cell viability experiments, EGFR P644S phenocopied EGFR WT, while the other ECD variants displayed erlotinib-insensitivity after 10–14 d of erlotinib treatment. The EGFR intracellular domain (ICD) variants were also erlotinib-insensitive, though the EGFR T725M and V769M variants were slightly less erlotinib-insensitive compared to EGFR H773R and V774M (Fig. 3d-f).

Though we used 3 lung cancer cell lines with the same endogenous EGFR Exon 19 deletion, we observed varying levels of functionality in our assays. While the overall trends amongst the cell lines were consistent (observed variant-insensitivity to erlotinib), the variations in the exact dose-response curves between cell lines may reflect variability in endogenous EGFR expression and activity, variation in exogenous EGFR expression, and other cellular factors, making clinical translation of these in vitro dose-response likely to be imprecise. Furthermore, the variation between phospho-ERK modulation, phospho-EGFR modulation, and TKI impact on cell survival is highly variable, which has long been known, where in one of many such examples, the relative impact of erlotinib on survival is more pronounced in HCC827 than in NCI-H1650 cells while the impact of erlotinib on phospho-EGFR is conversely more pronounced in NCI-H1650 cells than in HCC827 cells[42].

Next, we investigated the changes in signal transduction associated with EGFR variant erlotinib-insensitivity. We overexpressed the tested panel of EGFR extracellular and intracellular domain variants in PC9, HCC4006, and HCC827 cells and observed higher levels of total EGFR expression compared to endogenous EGFR in each cell line. After 24 h of erlotinib treatment, we observed a significant decrease in the levels of phospho-EGFR compared to the vehicle control, except in cell lines expressing T790M. Phospho-EGFR levels did not appear to correlate with erlotinib-insensitivity growth assays; however, phospho-ERK levels did appear to correlate with the ability of EGFR ECD variants to grow in the presence of erlotinib (Fig. 3g-l and Supp. Fig. 3a, b). We also assessed changes to phospho-AKT, -STAT1, and -STAT3, and did not observe changes correlating with growth in the presence of erlotinib compared to the controls (Supp. Fig. 3c-e). This was also true for the EGFR ICD variants tested, where phospho-ERK, but not phospho-EGFR, -AKT, -STAT1, or -STAT3, appeared to correlate with growth in the presence of erlotinib (Supp. Fig. 3f-h).

Since we observed an increase in the levels of phospho-ERK in the presence of erlotinib in cells expressing the tested EGFR variants, we sought to determine whether our panel of erlotinib-insensitive variants is dependent on MAPK signaling for growth. We treated our panel of EGFR VUS expressed in either PC9 or HCC4006 cells with trametinib, a well-characterized MEK inhibitor, alone and in combination with erlotinib. As expected, trametinib alone had no effect on colony formation, though when combined with erlotinib, a significant reduction

in colony formation was observed (Supp. Fig. 3i, j). Together, these data suggest that phospho-ERK levels rebound in the erlotinib-insensitive variants, which allow them to grow in the presence of erlotinib. Erlotinib-insensitivity is at least partially dependent on MAPK signaling as determined by suppression of colony formation after combination treatment with trametinib and erlotinib.

## Effects of osimertinib on EGFR intracellular domain variants in lung cancer models

Since 2018, osimertinib has been the standard of care for patients who present with EGFR-mutated lung cancer[43]. Given this, we sought to understand the effects of osimertinib treatment on both EGFR intracellular domain VKS and VUS in the PC9 cell line model (Fig. 4a), representing a subset of variants tested for erlotinib sensitivity and signaling above. First, we assessed a subset of EGFR VKS (E709A, G719A, L747P, S768I, G779F, L858R, and L861K) for cell viability and colony formation. EGFR G719A, S768I, and L858R were osimertinib-sensitive compared to the controls, while EGFR E709A, L747P, G779F, and L861K were osimertinib-insensitive (Fig. 4b, c, e). Osimertinib is known to specifically target EGFR-containing kinase domain mutations, with particular potency for exon 19 deletion and L858R hotspots, while both EGFR WT and EGFR C797S variants are either osimertinib-insensitive or -partially insensitive at the concentrations employed[13,44]. Next, we assessed changes to downstream signaling, finding that osimertinib-sensitivity appeared to correlate with a reduction in the levels of phospho-EGFR and phospho-ERK (at higher doses), but not phospho-AKT, -STAT1, or -STAT3 after 24 h of osimertinib treatment (Fig. 4d, f, g and Supp. Fig. 4a).

Next, we assessed osimertinib sensitivity in our panel of EGFR intracellular domain VUS (T725M, V769M, H773R, and V774M). We observed osimertinib-insensitivity at low doses, but osimertinib-sensitivity at higher doses for a subset of variants, in a cell viability and colony formation assay in both PC9 and HCC827 cell lines (Fig. 4h–j and Supp. Fig. 4b, c). To study the downstream signaling consequences of osimertinib treatment for the tested kinase domain variants, we assessed changes to phospho-EGFR, -ERK, -AKT, -STAT1, and -STAT3 in the PC9 cell line at varying osimertinib doses (Fig. 4k-m and Supp. Fig. 4d, e). We observed a reduction in the levels of phospho-EGFR and -ERK at higher doses, which was consistent with our observed growth phenotypes. Based on this we, we hypothesized that changes in MAPK signaling could be mediating dependency. We treated PC9 and HCC827 with trametinib, alone and in combination with osimertinib, and found a reduction in colony formation with the combination suggesting significant MAPK-dependency (Supp. Fig. 4f, g), as shown above for the tested EGFR variants.

## Effects of afatinib and dacomitinib on EGFR intracellular domain variants in lung cancer models

Next, we sought to understand whether our panel of kinase variants were functional in the presence of the second-generation EGFR TKIs, afatinib and dacomitinib. First, we assessed the same subset of EGFR VKS (E709A, G719A, L747P, S768I, G779F, L858R, and L861K) for cell viability and colony formation. EGFR G719A, S768I, and L858R were afatinib- and dacomitinib-sensitive compared to the controls, while EGFR E709A, L747P, G779F, and L861K were less sensitive than the controls in the cell viability assay (Fig. 5a,b). In the colony formation assay, higher doses of either afatinib or dacomitinib resulted in reduced growth for all tested VKS beyond T790M, which is known to be insensitive (Supp. Fig. 5a, b). We found levels of phospho-ERK treated with varying doses of afatinib or dacomitinib in PC9 cells correlated with the ability to grow in the presence of either drug (Supp. Fig. 5c-f). These data suggest that all of the VKS variants tested were inhibited in the presence of high doses of either afatinib or dacomitinib.

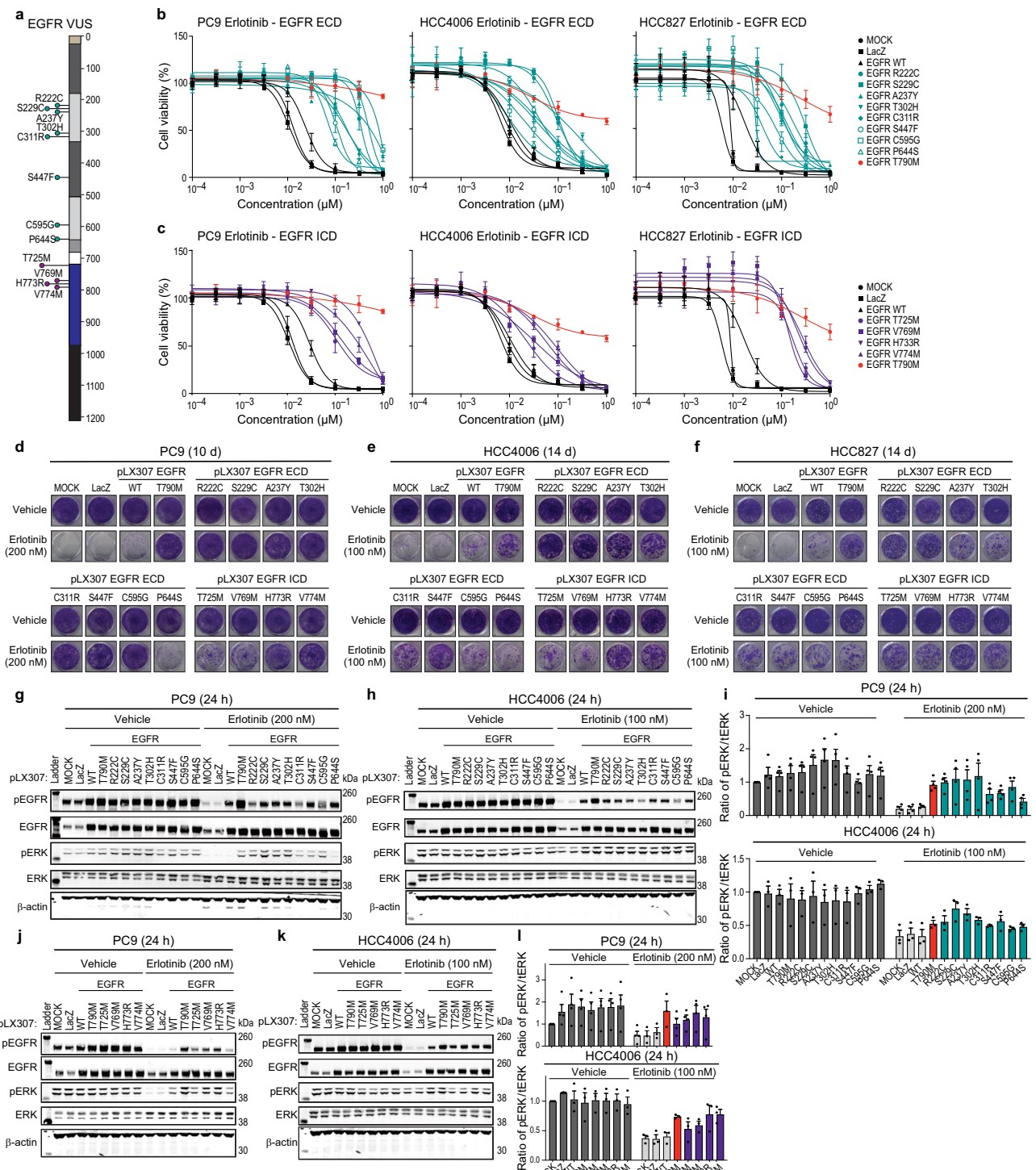

**Fig. 3 | Characterization of EGFR variants of unknown significance (VUS) in lung cancer models of EGFR addiction. a** EGFR domain schematic showing tested variants of unknown significance (VUS). **b, c** PC9, HCC4006, or HCC827 cell lines expressing either LacZ, EGFR WT, EGFR T790M, EGFR extracellular domain variants (R222C, S229C, A237Y, T302H, C311R, S447F, C595G, or P644S) (**b**), or EGFR intracellular domain variants (T725M, V769M, H773R, or V774M) (**c**) after 144 h treatment with increasing doses of erlotinib (normalized to vehicle control). A representative experiment is shown, remaining biological replicates are located in Source Data (N = 3). Data are presented as mean values +/- SD (**b, c**). **d-f** Colony formation with PC9 EGFR mutants and controls as in (**b**) and (**c**) after 10 d (PC9) or 14 d (HCC4006 and HCC827) of treatment with either vehicle (DMSO), 100 nM erlotinib (HCC4006 or HCC827), or 200 nM erlotinib (PC9). A representative experiment is shown (N = 3). **g-l** Representative immunoblots displaying the effect of either 200 nM (PC9) or 100 nM (HCC4006) erlotinib after 24 h treatment of cell lines expressing either LacZ, EGFR WT, EGFR T790M, EGFR extracellular domain variants (R222C, S229C, A237Y, T302H, C311R, S447F, C595G, or P644S) (**g, h, i**), or EGFR intracellular domain variants (T725M, V769M, H773R, and V774M) (**j, k, l**) on the levels of both phosphorylated EGFR and ERK and total EGFR and ERK. β-actin immunoblotting was used to determine equivalent loading. Data are presented as mean values ± SEM of biological replicates (PC9, N = 4; HCC4006, N = 3) (**i, l**).

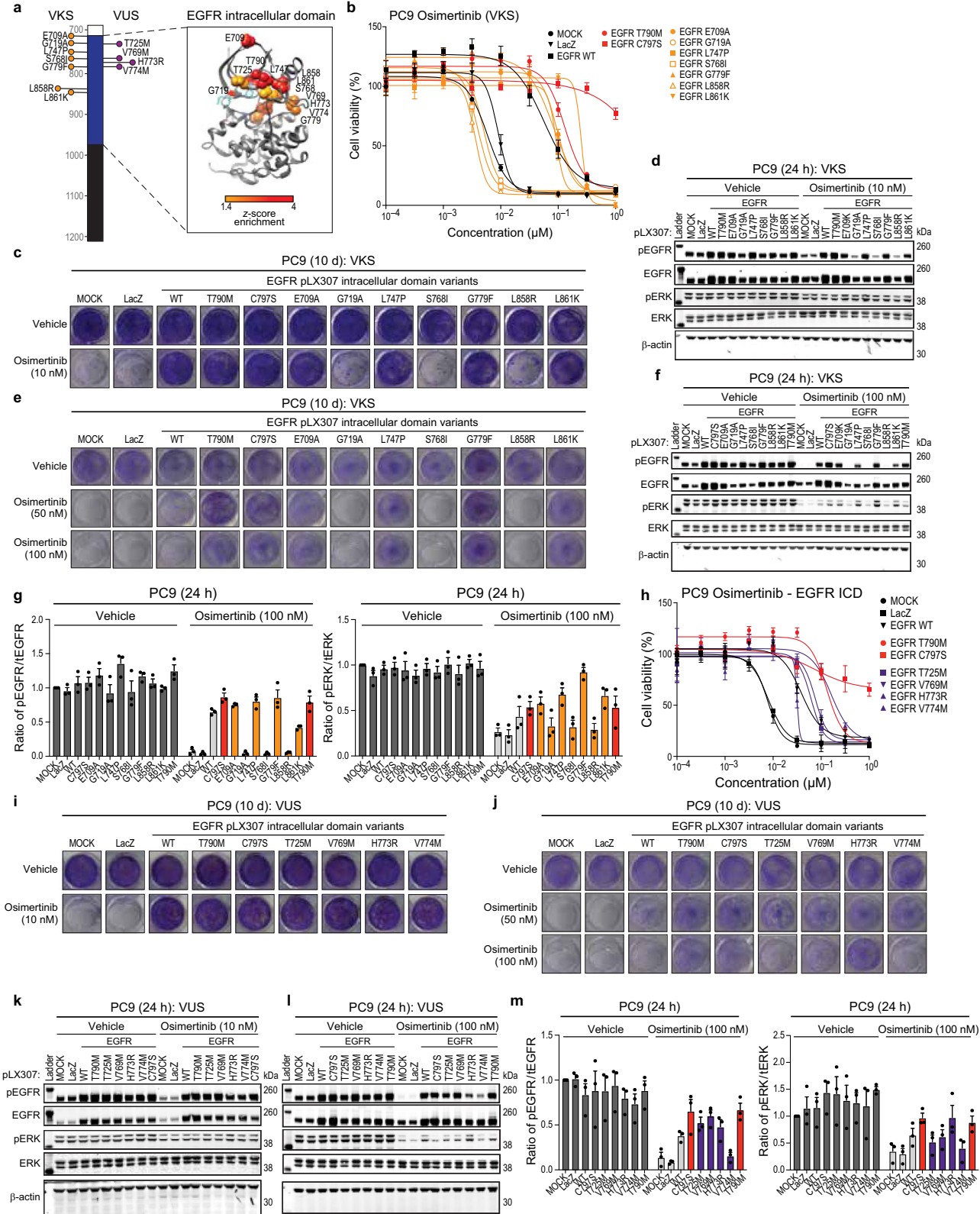

Next, we assessed functionality in our panel of EGFR kinase domain VUS (T725M, V769M, H773R, and V774M), in the presence of either afatinib or dacomitinib. Cells expressing H773R and V774M mutants are less sensitive to afatinib or dacomitinib than those expressing T725M or V769M mutants (Fig. 5c,d). In colony formation assays, EGFR H773R and V774M support lung cancer cell line growth in the presence of lower doses of afatinib and dacomitinib (Fig. 5e, f and

Supp. Fig. 5g,h) but are more inhibited at higher doses of afatinib and especially dacomitinib (Supp. Fig. 5h). These findings are consistent with the observed increase in levels of phospho-ERK, but not phospho-EGFR, -AKT, -STAT1, and -STAT3 for both EGFR H773R and V774M in the presence of afatinib (Supp. Fig. 5i-n). In the presence of high doses of dacomitinib, phospho-ERK levels are also suppressed (Supp. Fig. 5j). Together these data support the observation that treatment with high

**Fig. 4 | The effects of osimertinib treatment on EGFR intracellular domain variants. a** EGFR domain schematic and ribbon structure of both EGFR intracellular domain VKS and VUS. **b** PC9 cell lines expressing either LacZ, EGFR WT, EGFR T790M (controls), or EGFR VKS intracellular domain variants after 144 h treatment with increasing doses of osimertinib. A representative experiment is shown, remaining biological replicates are located in Source Data ($N$ = 3). Data are presented as mean values ± SD. **c** Colony formation with PC9 EGFR mutants and controls after 10 d of 10 nM osimertinib treatment. A representative experiment is shown ($N$ = 3). **d** Representative immunoblots displaying the effect of 10 nM osimertinib after 24 h treatment on PC9 EGFR mutants and controls on the levels of phosphorylated EGFR and ERK and total EGFR and ERK. β-actin is the loading control. **e** Colony formation with PC9 EGFR mutants and controls after 10 d of 50 or 100 nM osimertinib treatment. A representative experiment is shown (N = 3). **f, g** Representative immunoblots displaying the effect of 100 (**f**) nM osimertinib after 24 h treatment on PC9 EGFR mutants and controls on the levels of phosphorylated EGFR and ERK and total EGFR and ERK. β-actin is the loading control. Total levels of phosphorylated-EGFR or ERK were normalized to total levels of EGFR or ERK. Presented data are mean values ± SEM of biological replicates (N = 3) (**g**). **h** PC9 cell lines expressing controls or EGFR VUS intracellular domain variants after 144 h treatment with increasing doses of osimertinib. A representative experiment is shown, remaining biological replicates are located in Source Data ($N$ = 3). Data are presented as mean values ± SD. **i, j** Colony formation with PC9 EGFR mutants and controls after 10 d of 10, 50 or 100 nM osimertinib treatment. A representative experiment is shown ($N$ = 3). **k–m** Representative immunoblots displaying the effect of 10 (**k**) or 100 (**l**) nM osimertinib after 24 h treatment on PC9 EGFR mutants and controls on the levels of phosphorylated EGFR and ERK and total EGFR and ERK. β-actin is the loading control. Total levels of phosphorylated-EGFR or ERK were normalized to total levels of EGFR or ERK. Presented data are mean values ± SEM of biological replicates ($N$ = 3) (**m**).

doses of either afatinib or dacomitinib prevents or reduces activity for tested variants in the kinase domain. As mentioned above, given the differences between cell lines in the ectopic expression assays, we do not propose that these dose-response relationships apply directly to cancers harboring these mutants.

## Effects of afatinib and dacomitinib on EGFR extracellular domain VUS

Second generation EGFR TKIs have demonstrated some success in patients presenting with lung cancers with uncommon EGFR kinase domain mutations[18,45]. Given this, we sought to understand whether lung cancer cells expressing our panel of uncharacterized EGFR extracellular domain variants could have functionality in the presence of afatinib or dacomitinib. Since we identified a subset of uncharacterized EGFR extracellular domain variants (R222C, S229C, A237Y, T302H, C311R, S447F, C595G, and P644S) that were not functional in the presence of erlotinib, we sought to assess their functionality in the presence of afatinib and dacomitinib (Fig. 6a). First, we expressed our panel of EGFR ECD variants in PC9 and assessed cell viability in the presence of increasing doses of either afatinib or dacomitinib. We observed that variants treated with afatinib were less active compared to EGFR WT, except EGFR P644S, yet were not as active as EGFR T790M (Fig. 6b). When treated with dacomitinib the variants, except EGFR R222C, S229C, and T302H, were similarly as active as EGFR WT (Fig. 6c). We also expressed our EGFR ECD panel in HCC827, and assessed cell viability, observing functionality for all variants either equal to or greater than EGFR WT, except for EGFR R222C, S229C, and A237Y, though EGFR R222C, S229C, and A237Y were not as nearly as active as EGFR T790M (Supp. Fig. 6a, b).

Next, we assessed colony formation in PC9 cell lines, finding prolonged treatment with low doses afatinib led to a reduction in colony formation for EGFR C311R, C595G, and P644S, while for dacomitinib a reduction in colony formation was observed for EGFR R222C, A237Y, C311R, S447F, C595G, and P644S (Fig. 6d). In HCC827 cell lines, we observed a reduction in colony formation for EGFR R222C, T302H, C311R, S447F, C595G, and P644S after either afatinib or dacomitinib treatment (Supp. Fig. 6c). At higher doses in the PC9 cell line only EGFR S229C and T302H demonstrated activity in the presence of afatinib, while dacominitib prevented activity for all variants tested (Fig. 6e). Given these differences, we assessed changes to downstream signaling after 24 h of afatinib or dacomitinib treatment in PC9 cells and observed a rebound in the levels of phospho-ERK, but not phospho-EGFR, -AKT, -STAT1, or -STAT3, which appeared to correlate with growth at varying concentrations in the colony formation assay (Fig. 6f–i and Supp. Fig. 6e–j). To test the hypothesis that EGFR extracellular domain variants are dependent on MAPK pathway signaling in the presence of either afatinib or dacomitinib, we treated PC9 and HCC827 cell lines with either trametinib, alone or in combination with afatinib or dacomitinib. We observed reduced colony formation in the presence of the combination only, suggesting that activity in the presence of afatinib- and dacomitinib is dependent on MAPK pathway signaling (Supp. Fig. 6k, l). Together, these data have revealed a subset of EGFR extracellular domain variants lack activity in the presence of 2nd generation EGFR inhibitors in preclinical models.

## Treatment of glioblastoma harboring EGFR G598V mutation with dacomitinib

Based on the findings above, we sought to assess whether EGFR extracellular domain VKS would also be inhibited by 2nd generation EGFR TKIs. We expressed a subset of EGFR extracellular VKS in PC9 cells and assessed activity in the presence of either afatinib or dacomitinib in a cell viability assay. We observed this subset of variants to be more active compared to EGFR WT, with EGFR M277E being the most active and G598V being the least active of the variants tested after afatinib treatment (Supp. Fig. 7a). This observation was also true for PC9 cell lines treated with dacomitinib (Fig. 7a). Next, we assessed colony formation after low dose treatment with afatinib, finding only G598V demonstrated reduced activity, while the other variants were active. However, higher doses of afatinib reduced activity for all tested variants, except M277E (Supp. Fig. 7b). After high dose dacomitinib treatment, we observed reduced activity for EGFR R108K, M277E, A289V, and G598V (Fig. 7b and Supp. Fig. 7c). Functionality in the presence of either afatinib or dacomitinib in the colony formation assay appeared to correlate with the levels of phospho-ERK, not phospho-EGFR, upon treatment with either inhibitor (Fig. 7c,d and Supp. Fig. 7d–f). As expected, the EGFR extracellular domain VKS were functional in the presence of osimertinib (Supp. Fig. 7g, h).

Currently the standard of care for lung cancer patients presenting with EGFR extracellular domain variants is chemotherapy. We are not aware of any clinical studies assessing the efficacy of either afatinib or dacomitinib in lung cancer patients presenting with EGFR extracellular domain variants. However, this is not the case for patients presenting with *EGFR*-mutant glioblastoma, where clinical trials have assessed the efficacy of dacomitinib on extracellular domain variants, as well as other EGFR alterations[46,47]. Here, we show one such case, where a patient harboring an EGFR G598V mutation, received dacomitinib which led to disease stabilization (Fig. 7e). There are additional case reports of EGFR G598V glioblastoma, which also responded to dacomitinib, such that the tumor had a complete response[19]. In one previously reported case[19], we obtained sequence data for a subsequent relapse specimen. In this case, the relapse specimen exhibited two *NF1* loss of function mutations, Y2285* and R1276fs*9, but no longer harbored an *EGFR* mutation (Table 1)[19]. This result suggests that a heterogeneous glioblastoma population, containing *NF1* mutation, was selected via dacomitinib treatment. Together, these data support further investigations into dacomitinib sensitivity in cancers harboring EGFR ECD mutations.

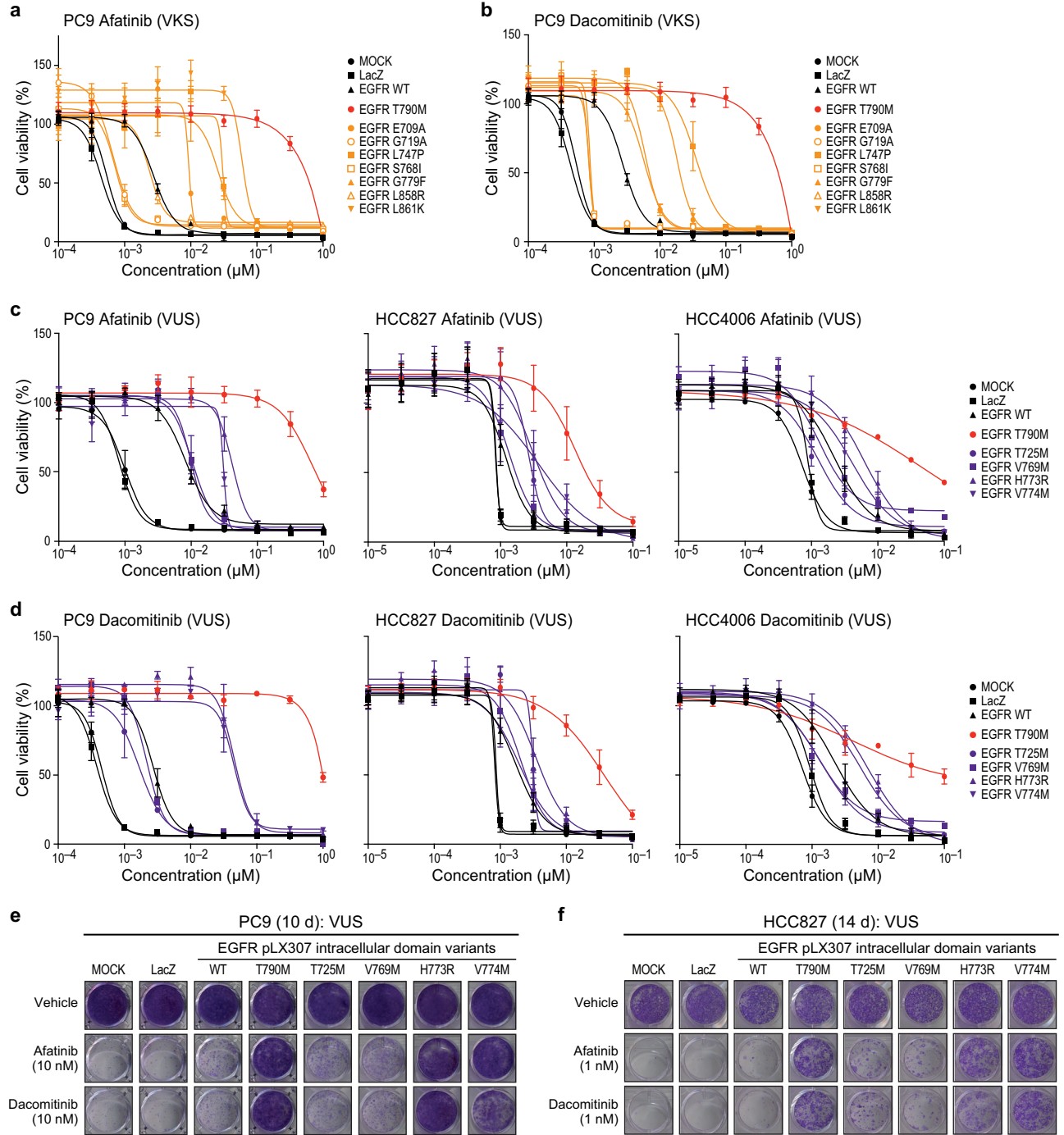

**Fig. 5 | EGFR intracellular domain variant sensitivity to afatinib and dacomitinib. a**, **b** PC9 cell lines expressing either LacZ, EGFR WT, EGFR T790M, or EGFR VKS intracellular domain variants (E709A, G719A, L747P, S768I, G779F, L858R or L861K) after 144 h treatment with increasing doses of afatinib (**a**) or dacominitib (**b**) in (normalized to vehicle control). A representative experiment is shown, remaining biological replicates are located in Source Data (N = 3). Data are presented as mean values ± SD. **c**, **d** PC9 cell lines expressing either LacZ, EGFR WT, EGFR T790M, EGFR extracellular domain variants (T725M, V769M, H773R, and V774M)

after 144 h treatment with increasing doses of afatinib (**c**) or dacomitinib (**d**) (normalized to vehicle control). A representative experiment is shown, remaining biological replicates are located in Source Data (N = 3). Data are presented as mean values ± SD. **e**, **f** Colony formation with either PC9 (**e**) or HCC827 (**f**) EGFR mutants and controls as in (**c**) and (**d**) after 10 or 14 d of treatment with either vehicle (DMSO), 1 nM, 10 nM, of either afatinib or dacomitinib. A representative experiment is shown (N = 3).

## Discussion

Building on previous studies[20–22,38] and applying new technical approaches of synthetic saturation mutagenesis[28], we report results from a comprehensive variant analysis of full-length EGFR in models of lung cancer. To date, *EGFR* is the largest gene to successfully undergo deep full-length mutational scanning, to our knowledge. To model

EGFR variant functionality, we employed the PC9 cell line, as it is EGFR-dependent, sensitive to all generations of EGFR TKIs, and amenable to high-throughput genetic screening, while also providing the complexity associated with naturally occurring passenger mutations in cancer. The PC9 cell line has also been shown to express TGFalpha, which can activate EGFR, though we believe pharmacological EGFR

inhibition would overcome any autocrine activation of EGFR in this context[48]. To validate the significance of pharmacological inhibition, we used additional *EGFR*-mutant lung cancer cell line models and observed similar phenotypes upon EGFR TKI treatment. We designed

and generated an EGFR variant overexpression library and assessed the ability of variants to grow in the presence of erlotinib. A benefit of the methodology used here is the pooled complexity and subsequent analyses which allowed for high-throughput screening of almost all

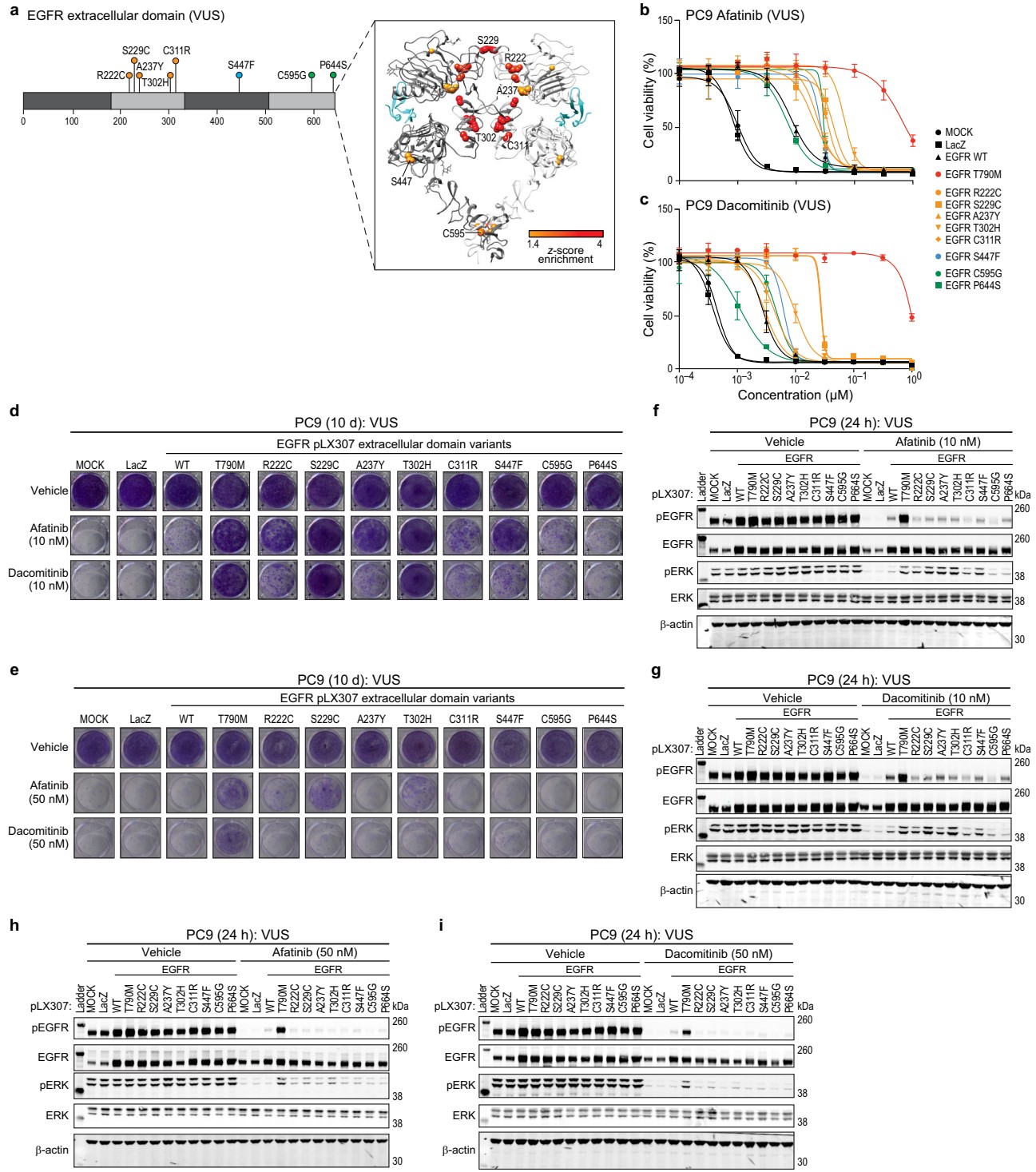

**Fig. 6 | EGFR extracellular domain variant sensitivity to afatinib and dacomitinib. a** Domain schematic and ribbon structure of EGFR extracellular domain variants of unknown significance (VUS). **b, c** PC9 cell lines expressing either LacZ, EGFR WT, EGFR T790M, EGFR extracellular domain variants (R222C, S229C, A237Y, T302H, C311R, S447F, C595G, or P644S) after 144 h treatment with increasing doses of afatinib (**b**) or dacomitinib (**c**) (normalized to vehicle control). A representative experiment is shown, remaining biological replicates are located in Source Data (N = 3). Data are

presented as mean values ± SD. **d, e** Colony formation with PC9 EGFR mutants and controls as in (**b**) and (**c**) after 10 d of treatment with either vehicle (DMSO), 10 nM (**d**) or 50 nM (**e**) of either afatinib or dacomitinib. A representative experiment is shown (N = 3). **f–i** Representative immunoblots displaying the effect of either afatinib (**f, h**) or dacominitib (**g, i**) after 24 h treatment on PC9 EGFR mutants and controls as in (**b**) and (**c**) on the levels of both phosphorylated EGFR and ERK and total EGFR and ERK. β-actin immunoblotting was used to determine equivalent loading.

possible variants in EGFR in a single pass. For cloning reasons, this approach does not generate a full saturation mutagenesis library, although the constructed library represents > 99% of all possible EGFR

substitution variants. Another caveat to the system is its use of EGFR overexpression. However, a method to comprehensively generate and introduce protein variants using CRISPR/Cas9 and base editing

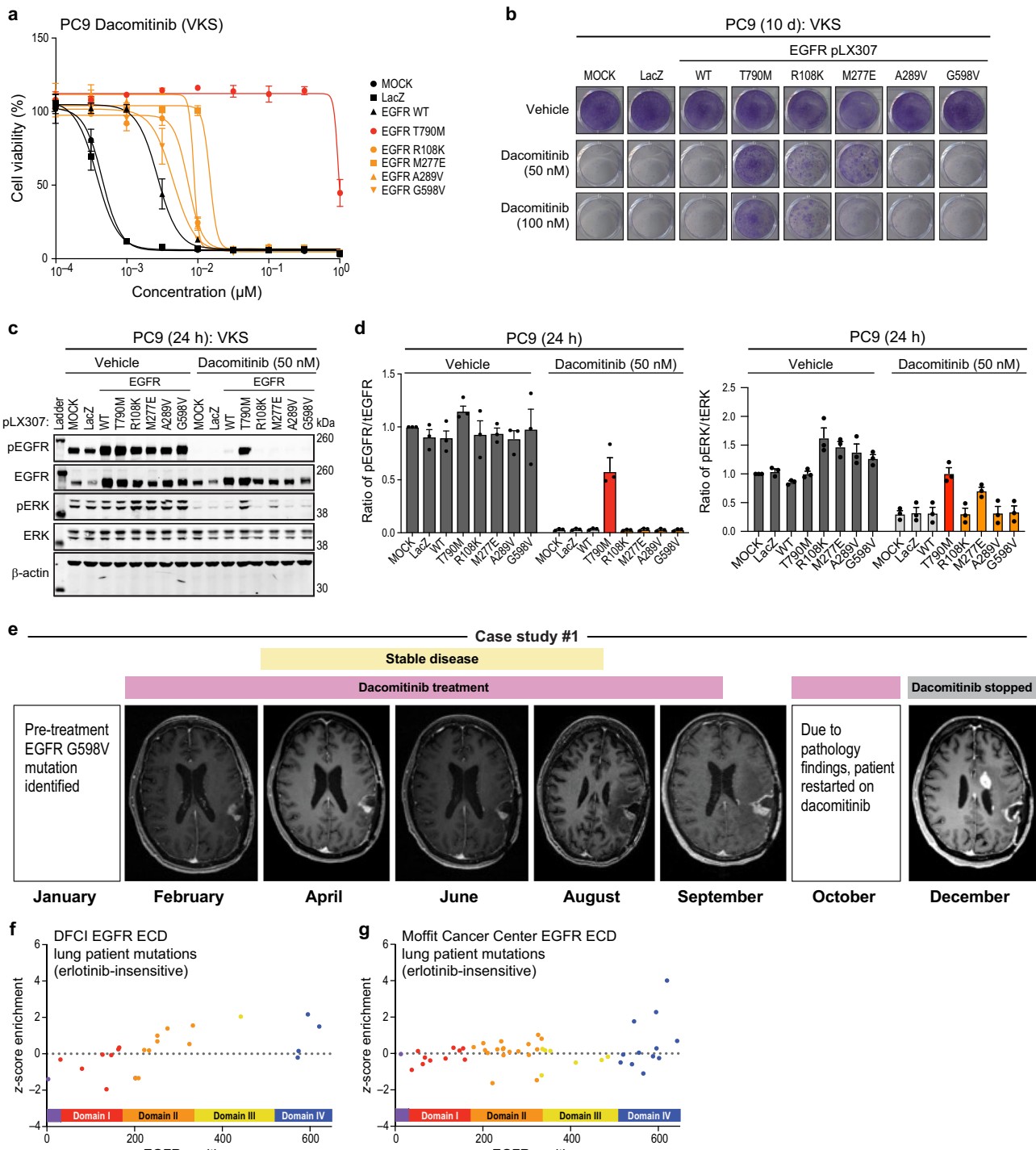

**Fig. 7 | Treatment of glioblastoma harboring EGFR G598V mutation with dacomitinib. a** PC9 cell lines expressing either LacZ, EGFR WT, EGFR T790M, EGFR VKS extracellular domain variants (R108K, M277E, A289V, or G598V) after 144 h treatment with increasing doses of dacomitinib (normalized to vehicle control). A representative experiment is shown, remaining biological replicates are located in Source Data (N = 3). Data are presented as mean values ± SD. **b** Colony formation with PC9 EGFR mutants and controls as in (**a**) after 10 d of treatment with either vehicle (DMSO), 50 nM, or 100 nM dacomitinib. A representative experiment is shown (N = 3). **c, d** Representative immunoblots displaying the effect of 10 nM

dacomitinib after 24 h treatment on PC9 EGFR mutants and controls as in (**a**) on the levels of both phosphorylated EGFR and ERK and total EGFR and ERK. β-actin immunoblotting was used to determine equivalent loading (**c**). Total levels of phosphorylated-EGFR or ERK were normalized to total levels of either EGFR or ERK. Data are presented as mean values ± SEM of biological replicates (N = 3) (**d**). **e** Images of an EGFR G598V positive glioblastoma patient after treatment with dacomitinib. **f, g** EGFR extracellular domain mutations observed in lung cancer patients from Dana-Farber Cancer Institute (**f**) or Moffit Cancer Center (**g**) based on z-score enrichment from the screen.

**Table 1 | Containing key alterations from a glioblastoma patient pre- and post-dacomitinib treatment**

| Case study #2 | |
| --- | --- |
| Glioblastoma patient: key genomic alterations (pre- and post-dacomitinib treatment) | |
| Pre-treatment | Post-treatment |
| EGFR G598V | NF1 Y2295* |
| CDKN2A/B loss | NF1 R1276fs*9 |
| CDKN2C L58fs*7 | CDKN2A/B loss |
| | CDKN2C L58fs*7 |

technology has yet to be developed, though this area of investigation is active[49].

The saturation mutagenesis screen revealed a number of functional EGFR variants spanning the length of EGFR, with an enrichment of EGFR erlotinib-insensitive variants observed in the dimerization, transmembrane, and kinase domains. Our data have significantly expanded the universe of known functional EGFR variants. However, one limitation of our study is that we cannot generate sequence-based or location-based rules that enable us to predict accurately the function of novel variants; this analysis thus remains an empirical exercise. Another limitation of the PC9 oncogene addiction model is the assay design, as a subset of functional variants have been reported, for example, G719 and L858, that could not be selected for under the assay conditions because these variants, while activating, are not erlotinib resistant. Moreover, the screen yielded a number of EGFR variants that have not been reported in patients. Subsets of these variants are unlikely to be reported because a di- or tri-nucleotide change would be required to generate them, which is unlikely to naturally occur. Though variants resulting from mono-nucleotide changes have a stronger probability of occurring, there is still a possibility that they will never naturally arise.

We observed tested EGFR ECD variants to be both erlotinib- and osimertinib-insensitive, so that other agents are needed for treatment of cancers bearing these variants. At high doses, the tested ECD variants were sensitive to the second-generation EGFR TKIs, afatinib and dacomitinib in multiple lung cancer cell lines, with particular sensitivity to dacomitinib under the tested conditions. While our experimental conditions do not enable the precise determination of the most effective inhibitor for a particular EGFR variant, they establish the general principle of EGFR ECD variant sensitivity to second-generation inhibitors. Since lung cancer patients harboring EGFR ECD variants are not usually considered as candidates for EGFR TKI therapy, we were unable to find literature reports assessing the effects of EGFR-targeted TKIs in such patients.

EGFR ECD variants are most common in glioblastoma, where clinical trials of EGFR TKI's have been reported. In aggregate, these clinical trials have not reported benefit for patients treated with dacomitinib[19,46,50] or other EGFR TKI's. Here, we provided data from one glioblastoma patient who presented with an EGFR G598V mutation, where dacomitinib treatment led to disease stabilization. A second patient whose glioblastoma harbored this mutation has previously been reported to have responded to dacomitinib[19]. Here, we report that this patient relapsed with a glioblastoma displaying compound loss of function mutations in *NF1*, suggesting that the dacomitinib was acting on target, inhibiting EGFR and selecting for EGFR-negative subclones. It is worth noting, however, that these clinical studies did not detect an overall association between the presence of EGFR ECD variants in archival tissue and response to dacomitinib. More specifically, EGFR ECD mutations were detected in 4 out of 8 patients who responded and 9 out of 23 patients who did not. The overall discrepancy between our pre-clinical data and the reported clinical trials could be a consequence of glioblastoma heterogeneity, of the rapid

development of resistance, as well as the sub-clonal nature of EGFR mutations in glioblastoma. For instance, it is possible that non-responding tumors with EGFR ECD mutations harbored the mutation in a small fraction of tumor cells, and that responding tumors where EGFR ECD mutations were not detected harbored these alterations in a part of the tumor that was not surgically sampled. Taken together, our report of the strong sensitivity of EGFR ECD variants to dacomitinib, as well as observed clinical responses of EGFR G598V, prompt additional consideration of dacomitinib for glioblastomas with EGFR ECD mutations. Additionally, we suggest that additional pre-clinical studies be initiated to evaluate the efficacy of dacomitinib as a possible therapeutic for *EGFR*-mutant lung cancer patients harboring ECD mutations, which are routinely observed as lung cancer somatic mutations (Fig. 7f,g). Though EGFR ECD variants make up less than 1% of total *EGFR*-mutant lung cancer cases every year[51], such cases could be evaluated for potential benefit from treatment with dacomitinib.

On a broader level, advances in technology have allowed us to create deep mutational scanning libraries to map variant functionality comprehensively and prospectively across protein families. This approach, matched with targeted panel DNA sequencing of patients, will enable us to determine underlying functionality and potential targetability of somatic alterations. There are numerous applications for this type of approach, when studying EGFR variants (or other protein families), including, but not limited to, assessing drug inhibitor resistance, variant transformation, and structure/function analysis. Beyond single nucleotide alterations, there are numerous structural alterations reported in cancer patients. EGFR, specifically, is also characterized by in-frame insertions and deletions (indel), fusions, and truncations[6,7]. To date, systematic assessment of protein truncations, fusions, or indels rely solely on reported variants, but the approach reported here could also be used to support systematic evaluation of other classes of structural alterations. Furthermore, recent developments in structural analysis of full-length EGFR using cryo-electron microscopy[52] may synergize with systematic mutagenesis studies as shown here.

Together, our findings support the use of high-throughput genetic screening to understand EGFR variant functionality in models of lung cancer. From this approach we identified erlotinib-insensitive variants and suggest that a subset of ECD variants, lacking approved EGFR-targeted therapies, could be afatinib- and dacomitinib-sensitive. Future pre-clinical and possibly clinical studies will be needed to assess the use of dacomitinib or other EGFR TKI's as a treatment strategy for lung cancer patients, and potentially glioblastoma patients, whose cancers harbor EGFR ECD variants.

## Methods

### Ethics statement

Ethics approval was granted in accordance with Institutional Review Board-approved protocols at MGH. The patient was a participant in clinical trial NCT01112527 (PF-00299804 in Adult Patients with Relapsed/Recurrent Glioblastoma). Informed consent was obtained from the patient's family for the publication of any identifiable images or other information included in this article.

### Lentiviral screening expression vector

Lentiviral vector pMT_025 (RRID:Addgene, catalogue no. 158759) was developed by Broad Institute Genetic Perturbation Platform (GPP). Open Reading Frames (ORF) can be cloned in through restriction/ligation at a multiple cloning site (MCS). The ORF expression is driven by EF1a promoter. A PAC gene is driven by SV40 promotor to confer puromycin resistance.

### Library design and cloning

The full-length EGFR gene was mutagenized. At each codon position, except the start and stop codons, we tried to make 19 missense

changes, and 1 nonsense change. The library consists of ~24,000 variants. The cloning protocol was described in ref. 28. The mutagenesis library was synthesized by Twist BioScience, according to our designs. The library was delivered as a pool of linear fragments representing the full-length EGFR ORF with a short flank sequence at each end. The two flank sequences were designed to facilitate restriction/ligation cloning of the linear fragment library into pMT_025 expression vector. The linear fragment library and the vector were each digested with NheI and MluI, and then ligated. The ligation products were then transformed into Stbl4 competent cells (New England BioLabs, catalogue no. C3040H). We obtained ~24 million colonies, meaning, on average, each variant got 1000 colonies. The colonies were harvested, and the plasmid DNA was extracted via maxi preparation kit (Qiagen, catalogue no. 12165). The resulting pDNA library was sequenced via Illumina Nextera XT platform. The distribution of variants was assessed.

### Lentivirus production for screen

Lentivirus was created by Broad Institute GPP. The detailed protocol is available at http://www.broadinstitute.org/rnai/public/resources/protocols/. Briefly, viral packaging cells 293T was transfected with pDNA library, a packaging plasmid containing gag, pol and rev genes (e.g. psPAX2, RRID:Addgene, catalogue no. 12260), and VSV-G expressing envelop plasmid (e.g. pMD2.G, RRID:Addgene, catalogue no. 12259), using TransIT-LT1 transfection reagent (Mirus Bio, catalogue no. MIR2300). Media was changed 6–8 h post-transfection. Virus was harvested 30 h post-transfection.

### EGFR saturated mutagenesis screen

PC9 cells were infected with the lentiviral EGFR mutant library overexpressing ~24,000 variants at low MOI. Approximately 48 h after infection cells were selected with puromycin for 3-5 days to remove uninfected cells. After selection, cells were split and treated with 200 nM erlotinib. Cells were passaged in drug or fresh media (untreated arm) every 3-4 days. Cells were harvested 10 days after initiation of treatment. Genomic DNA was isolated according to the manufacturer 's protocol (Qiagen).

### PCR for screen deconvolution

The integrated ORF in genomic DNA was amplified by PCR. The PCR products were shotgun sheared with transposon, index labeled, and sequenced with next-generation sequencing technology. The general screen deconvolution strategy and considerations were described in detail in Yang et al.[28]. The PCR primers were designed in such a way that there is a ~100 bp extra sequence at each end leading up to the mutated ORF region. We used these 2 primers:

Forward: 5′- TGGCACTTGATGTAATTCTCCTTGGA −3′
Reverse: 5′- TTAAAGCAGCGTATCCACATAGCGT −3′
PCR cycling program:
1. 95 C 30 s
2. 98 C 10 s
3. 69 C 30 s (Annealing temperature when using Q5 Hot Start High-
Fidelity 2X Master Mix (New England BioLabs))
4. 72 C 2.5 min
5. Go to Step 2 34 times
6. 72 C 2 min
7. 4 C hold

A full 96-well PCR reaction was used for each gDNA sample. Each PCR reaction is in 50 uL, and with 250 ng gDNA. Q5 Hot Start High-Fidelity 2X Master Mix (New England BioLabs) was used as DNA polymerase. 1/3 of 96 PCR reactions of a gDNA sample were pooled, concentrated with Qiagen PCR cleanup kit, and then purified by 1% agarose gel. The excised bands were purified first by Qiagen Qiaquick kits, then by AMPure XP kit (Beckman Coulter).

### Nextera sequencing for screen deconvolution

Following Illumina Nextera XT protocol, for each sample 6 Nextera reactions were set up, each with 1 ng of purified ORF DNA. Each reaction was indexed with unique i7/i5 index pairs. After the limited-cycle PCR step, the Nextera reactions were purified with AMPure XP kit. All samples were then pooled and sequenced with Illumina Novaseq S4 platform.

### Analysis of Nextera sequencing data

NovaSeq S4 data were processed with software "AnalyzeSaturationMutagenesis" developed by Broad Institute[28]. Typically, the pair-end reads were aligned to reference sequence. Multiple filters were applied, and some reads were trimmed. The counts of detected variants were then tallied. The output files from AnalyzeSaturationMutagenesis, one for each screening sample, were then parsed, annotated merged into a single.cvs file that is ready for candidate analysis. These software tools are freely available, as described in ref. 28.

### Analysis of the variant screen

The log2(fold-change) in EGFR variant representation between cells treated with erlotinib for 10 days and the initial early time point of variant plasmid used to generate virus was calculated. Then a Z-score was calculated, and a variant was considered significant if the Z-score ≥ 1.5 (See Supplementary Data 1).

### Cell lines

The HCC4006 (catalogue no. CRL-2871) lung cancer cell line was purchased from ATCC, and PC9 and HCC827 lung cancer cell lines were obtained from the Cancer Cell Line Encyclopedia (CCLE; RRID:SCR_013836) and DepMap (Cancer Dependency Map Portal, RRID:SCR_017655). The sources for these lung cancer lines are listed at DepMap.org, and they can be obtained from their respective sources. Their identities were confirmed by single-nucleotide polymorphism array. Cell lines were confirmed negative for Mycoplasma infection (Lonza MycoAlert, catalogue no. LT07-318). Cell line identities were reconfirmed by short tandem repeat (STR) profiling at Dana-Farber Cancer Institute (Molecular Diagnostics Laboratory Core). Cells were maintained in RPMI-1640 (PC9, HCC827, and HCC4006) supplemented with 10% fetal bovine serum (Sigma) and incubated at 37 °C in 5% $CO_2$.

### Immunoblot blot reagents

Cells were lysed in RIPA Buffer (25 mM Tris•HCl pH 7.6, 150 mM NaCl, 1% NP-40, 1% sodium deoxycholate, 0.1% SDS; phosphatase and protease inhibitors) and resolved by Tris-Bis SDS-PAGE. To determine the levels of activated proteins, immunoblot analyses were done with phospho-specific antibodies to EGFR (Cell Signaling Technology, catalogue no. 3777), AKT (Cell Signaling Technology, catalogue no. 4060), MEK1/2 (Cell Signaling Technology, catalogue no. 9121), RB1 (Cell Signaling Technology, catalogue no. 8516), ERK1/2 (Cell Signaling Technology, catalogue no. 3470), STAT1 (Cell Signaling Technology, catalogue no. 9167), STAT3 (Cell Signaling Technology, catalogue no. 9145) and with antibodies recognizing total EGFR (Cell Signaling Technology, catalogue no. 2232), AKT (Cell Signaling Technology, catalogue no. 58295), RB (Cell Signaling Technology, catalogue no. 9309), MEK1/2 (Cell Signaling Technology, catalogue no. 4694), ERK1/2 (Cell Signaling Technology, catalogue no. 9102), CCNB1 (Cell Signaling Technology, catalogue no. 4138), STAT1 (Cell Signaling Technology, catalogue no. 14994), and STAT3 (Cell Signaling Technology, catalogue no. 9139) to control for total protein expression (Cell Signaling Technologies). Antibody for β-actin (Sigma, catalogue no. AC15) and Vinculin (EMD Millipore, catalogue no. 05-386) were used to verify equivalent loading of total cellular protein. Each immunoblot represents the results from three

independent lentiviral infections (N = 3). Please see Source Data file for all uncropped and unprocessed blots (Main and Supplementary Figs.).

## Western blot analyses

To analyze replicates of western blots, we performed densitometry using ImageJ 1.53k (http://imagej.nih.gov/ij). We used a standard ROI to determine the brightness of each band within the given ROI. After brightness was determined, the ratio of phospho-protein to total-protein was calculated.

## Western blot inhibitor concentrations

Afatinib and dacomitinib were used at 1 nM (HCC827 and HCC4006), 10 nM (PC9), or 50 nM (PC9) alone or in combination with trametinib. Erlotinib was used at either 100 nM (HCC827 and HCC2006) or 200 nM (PC9) alone or in combination with trametinib. Osimertinib was used at 10 nM (HCC827 and PC9), 50 nM (PC9), or 100 nM (PC9) alone or in combination with trametinib. Trametinib was used at either 1 nM (HCC827 and HCC4006) or 20 nM (PC9) alone or in combination with EGFR-TKIs.

## Small molecule inhibitors

Trametinib (catalogue no. S2673), erlotinib (catalogue no. S7786), afatinib (catalogue no. S1011), dacomitinib (catalogue no. S2727), and osimertinib (catalogue no. S7297) were purchased from Selleck Chemicals. Trametinib, erlotinib, gefitinib, afatinib, dacomitinib, and osimertinib were dissolved in DMSO and stored at stock concentrations of 10 mM (Trametinib, erlotinib, dacomitinib, and osimertinib) or 20 mM (gefitinib and afatinib) at −20 °C.

## Lentiviral expression vector infections

The pLX307 (RRID:Addgene, catalogue no. 41392) LacZ, EGFR (WT, T790M, C797S, R222C, S229C, A237Y, A289D, A289T, A289V, T302H, C311R, S447F, C595G, G598A, G598V, P644S, E709A, E709K, G719A, G719C, G719D, G719S, T725M, L747P, L747S, V769L, V769M, H773R, V774M, G779F, L858R, and L861K) puromycin lentivirus vector were cloned from a vector provided by the Broad Institute Genetic Perturbation Platform, and were transiently transfected into 293T cells with a Δ8.9 (RRID:Addgene, catalogue no. 132929) and VSV-G packaging system using Fugene6 (Promega, catalogue no. E2691). Infection of PC9, HCC827, and HCC4006 cell lines were performed in growth media supplemented with 5 µg/ml polybrene and selected with 2 µg/ml of puromycin for 72 h.

## Anchorage-dependent growth assays

To monitor proliferation, cells were plated into 96-well plates at a density of $2 \times 10^3$ (PC9, HCC827, and HCC4006) in triplicate. To quantitate cell number, after 6 days cells were stained with CellTiter-Glo (Promega, catalogue no. G7570) and luminescence was measured. We also performed a second proliferation assay to monitor clonogenic growth. Cells were plated at $5 \times 10^3$ (PC9) in 12-well plates or $5 \times 10^3$ (HCC827 or HCC4006) in 24-well plates in duplicate. After 10–14 days, paraformaldehyde-fixed cells were stained with crystal violet to visualize and quantitate colony growth.

## Determination of GI$_{50}$

To determine the 50% growth inhibitory concentration (GI$_{50}$), $2 \times 10^3$ cells were seeded in 96-well plates in triplicate and treated with concentrations of erlotinib, afatinib, dacomitinib, and osimertinib (from 3 nM to 1 µM) in the PC9 cell line. For HCC827 and HCC4006 we treated with concentrations of erlotinib from 3 nM to 1 µM and for afatinib and dacomitinib we treated from 1 nM to 100 nM. After 6 days cells were stained with CellTiter-Glo and luminescence was measured. GI$_{50}$ values were calculated with PRISM (RRID:SCR_005375) software.

## Statistical analysis and reproducibility

All statistical analyses for dose-response curves were performed with GraphPad 8.0 software (N = 3). Data are presented as means ± SD. Western blot experiments were repeated a minimum of three times, and representative images were shown. Quantification of western blots was performed using ImageJ 1.53k (http://imagej.nih.gov/ij). Data are presented as means ± SEM.

## Erlotinib titration for EGFR variant screen

The doses of erlotinib to use in these screens were determined by propagating cells in different concentrations of erlotinib to determine the effect on cell proliferation. In parallel, the level of phospho-ERK in cells treated with different concentrations of erlotinib was determined. For the proliferation assay, $1 \times 10^6$ PC9 cells were seeded in 10 cm plates without drug. PC9 cells were allowed to adhere for 24 h, after which fresh media containing different concentrations of erlotinib was added. PC9 cells were passaged, or media was refreshed every 3 days, and cells were counted at each passage. For immunoblots, PC9 cells were treated with DMSO or the indicated concentrations of erlotinib for 48 h. Phospho-ERK levels were assessed by immunoblot analysis.

## EGFR G598V case study

The patient was a participant in clinical trial NCT01112527 (PF-00299804 in Adult Patients with Relapsed/Recurrent Glioblastoma)[19]. Informed consent was obtained from the patient's family for the publication of any identifiable images or other information included in this article.

## Mapping of phenotypic data onto structures

Mapping and graphics of phenotypic data onto crystal structures were performed with the UCSF Chimera package[53]. Chimera is developed by the Resource for Biocomputing, Visualization, and Informatics at the University of California, San Francisco (supported by NIGMS P41-GM103311). Phenotypes were mapped using the "define and render by attribute" functions of the program with colors corresponding to z-scores of log-2-fold-changes of reads per mutant as indicated on the color key accompanying each picture. Mutated residues with phenotypes shown have their atoms/bonds selected as shown and illustrated as spheres.

## Reporting summary

Further information on research design is available in the Nature Portfolio Reporting Summary linked to this article.

# Data availability

Please see source data file for all uncropped and unprocessed blots (Main and Supplementary Figs.). EGFR comprehensive mutational scanning variant data are available within the Supplementary Information (Supplemental Table 1). Source data are provided with this paper.

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

## Acknowledgements

This work was supported by the Starr Cancer Consortium Grant (M.M.). We acknowledge additional support from the National Cancer Institute to T.K.H. (1K99CA248836) and M.M. (1R35CA197568), Burroughs Wellcome Fund Postdoc Enrichment Program Grant to T.K.H., and Damon Runyon Cancer Research Foundation award to T.K.H. (David Livingston Fellow). M.M. is an American Cancer Society Research Professor. We acknowledge L. Gaffney for figure design support.

## Author contributions

T.K.H.: Conceptualization, investigation, visualization, methodology, writing-original draft, writing-review, and editing. E.A.: Investigation, visualization, methodology, writing-review, and editing. N.S.P.: Investigation, visualization, methodology, writing-review, and editing. E.E.K.: Investigation, writing-review, and editing. L.B.: Investigation, writing-review, and editing. A.B.G.: Investigation, writing-review, and editing. D.A.: Investigation, writing-review, and editing. T.S.: Software, formal analysis, methodology, writing-review, and editing. R.E.S.: Formal analysis, writing-review, and editing. L.W.: Formal analysis, writing-review, and editing. L.G.: Investigation, writing-review, and editing. B.R.S.: Investigation, writing-review, and editing. M.D.M: Investigation, writing-review, and editing. T.R.F.: Investigation, writing-review, and editing. E.B.: Investigation, writing-review, and editing. Y.Y.L.: Formal analysis, writing-review, and editing. A.D.C.: Formal analysis, writing-review, and editing. F.P.: Investigation, writing-review, and editing. X.Y.: Software, formal analysis, methodology, writing-review, and editing. D.E.R.: Methodology, writing-review, and editing. J.K.H: Investigation, methodology, writing-review, and editing. D.P.C.: Investigation, visualization, methodology, writing-review, and editing. A.S.C.: Investigation, visualization, methodology, writing-review, and editing. J.D.: Investigation, visualization, methodology, writing-review, and editing. T.T.B.: Investigation, visualization, methodology, writing-review, and editing. C.M.J.: Conceptualization, funding acquisition, supervision, methodology, writing-review, and editing. M.M.: Conceptualization, funding acquisition, supervision, methodology, writing-original draft, writing-review, and editing.

## Competing interests

M.M. receives research support from Bayer Pharmaceuticals and Janssen. M.M. is a consultant for and equity holder in DelveBio, Interline and Isabl, and holds patents licensed to Bayer and LabCorp. C.M.J. is a full-time employee of and equity holder in Novartis. D.P.C. consults for the Massachusetts Institute of Technology, Advise Connect Inspire, German Accelerator, Lilly, GlaxoSmithKline, Incephalo, Boston Pharmaceuticals, Servier, Boston Scientific and Pyramid Biosciences (equity interest) for advisory input. He has also received financial compensation and travel reimbursement from Merck for invited lectures, and from the US NIH and DOD for clinical trial and grant review. F.P. is a full-time employee of Merck Research Laboratories. A.D.C. receives research support from Bayer Pharmaceuticals. D.E.R. receives research funding from members of the Functional Genomics Consortium (Abbvie, Bristol-Myers Squibb, Jannsen, Merck, Vir), and is a director of Addgene, Inc. A.S.C. is a full-time employee of and equity holder in Alterome Therapeutics. J.D. is a consultant for Novartis, Amgen, Janssen and Ono Therapeutics. N.S.P. is a full-time employee of Aera Therapeutics. The remaining authors declare no competing interests. J.K.H. serves as a consultant for Jackson Laboratory for Genomic Medicine.

## Additional information

[1]Department of Medical Oncology, Dana-Farber Cancer Institute & Harvard Medical School, Boston, MA, USA. [2]Cancer Program, The Broad Institute of M.I.T. and Harvard, Cambridge, MA, USA. [3]Genetic Perturbation Platform, The Broad Institute of M.I.T. and Harvard, Cambridge, MA, USA. [4]Data Science Platform, The Broad Institute of M.I.T. and Harvard Cambridge, Cambridge, MA, USA. [5]Summer Honors Undergraduate Research Program, Harvard Medical School, Boston, MA, USA. [6]Department of Pathology, H. Lee Moffitt Cancer Center and Research Institute, Tampa, FL, USA. [7]Center for Neuro-Oncology, Division of

Neuro-Oncology, Massachusetts General Hospital, Boston, MA, USA. [8]Department of Neurology, Division of Neuro-Oncology, Massachusetts General Hospital, Boston, MA, USA. [9]Department of Neurology, Brigham and Women's Hospital & Harvard Medical School, Boston, MA, USA. [10]Present address: Department of Molecular and Medical Pharmacology, University of California, Los Angeles, CA, USA. [11]Present address: Aera Therapeutics, Cambridge, MA, USA. [12]Present address: Merck Research Laboratories, Cambridge, MA, USA. [13]Present address: Department of Oncology, Novartis Institutes for Biomedical Research, Cambridge, MA, USA. ✉e-mail: matthew_meyerson@dfci.harvard.edu

