## [Peer Review File · Nature Communications]

Reviewers' Comments:

Reviewer #1:

Remarks to the Author:

The authors have addressed my minor concerns from the initial submission.

Reviewer #2:

Remarks to the Author:

This is an impressive and interesting study that goes well beyond previous screens of the effects of EGFR mutations on activity and inhibitor sensitivity, by using a deep mutational scanning library and thoroughly characterizing signaling and inhibitor sensitivity. It is a nice study, very well put together and presented, and should be published in *Nature Communications* in this reviewer's opinion – both because of the impressive use of the technology and because (as the authors state) they have 'significantly expanded the universe of known functional EGFR variants' with an excellent body of work.

There is one issue that the authors should discuss, though, that may influence interpretation of some of the results. There are old data (PMID 2783462) showing that PC9 cells express TGF α , which activates EGFR, suggesting that the PC9 experiments here are performed in the context of a degree of autocrine EGFR signaling. This is absolutely not a problem for the study, potentially making it even more interesting, with some mutations uncovered possibly sensitizing to ligand activation (which is very relevant) rather than just activating. The point should be discussed, though, in my opinion, and may provide an additional dimension to some of the interpretation of the data. Also, are there any similar data for the other cell lines used? Or might this explain some of the differences seen between cells?

Reviewer #3:

Remarks to the Author:

The authors have satisfactorily addressed all of the questions raised in my initial review. I believe this paper represents an important contribution to the literature and is appropriate for *Nature Communications*.